# Optimal Time Interval between Neoadjuvant Platinum-Based Chemotherapy and Interval Debulking Surgery in High-Grade Serous Ovarian Cancer

**DOI:** 10.3390/cancers15133519

**Published:** 2023-07-06

**Authors:** Angeliki Andrikopoulou, Charalampos Theofanakis, Christos Markellos, Maria Kaparelou, Konstantinos Koutsoukos, Kleoniki Apostolidou, Nikolaos Thomakos, Dimitrios Haidopoulos, Alexandros Rodolakis, Meletios-Athanasios Dimopoulos, Flora Zagouri, Michalis Liontos

**Affiliations:** 1Department of Clinical Therapeutics, Alexandra Hospital, Medical School, 11528 Athens, Greece; chrismarkellos@hotmail.com (C.M.); kapareloum@gmail.com (M.K.); koutsoukos.k@gmail.com (K.K.); apostolidoukl@gmail.com (K.A.); mdimop@med.uoa.gr (M.-A.D.); florazagouri@yahoo.co.uk (F.Z.); mliontos@gmail.com (M.L.); 21st Department of Obstetrics and Gynecology, Alexandra Hospital, Medical School, 11528 Athens, Greece; ch.theofanakis@gmail.com (C.T.); nthomakos@med.uoa.gr (N.T.); dimitrioshaidopoulos@gmail.com (D.H.); arodolak@med.uoa.gr (A.R.)

**Keywords:** ovarian cancer, IDS, time interval, cytoreductive surgery, neoadjuvant, progression-free survival

## Abstract

**Simple Summary:**

The optimal time interval between the completion of neoadjuvant chemotherapy (NACT) and interval debulking surgery (IDS) in high-grade serous ovarian carcinoma (HGSC) is not well defined. We conducted a retrospective study of patients with HGSC stage IIIC/IV who had received NACT followed by IDS during a 15-year period (January 2003–December 2018) in our Institution. Performing IDS within four weeks after NACT was associated with better survival outcomes. On multivariate analysis, the performance of IDS within four weeks after NACT was an independent factor of both PFS (*p* = 0.004) and OS (*p* = 0.003). Our study provides evidence that surgical intervention should not be significantly delayed after neoadjuvant chemotherapy.

**Highlights:**

**What are the main findings?**
The time interval NACT to IDS < 4 weeks was significantly associated with a prolonged PFS (*p* = 0.004) and OS (*p* = 0.002).Median OS was 66.3 months (95% CI: 39.1–93.4) vs. 39.4 months (95% CI: 31.8–47.0) in the <4 week vs. ≥4 week time interval NACT to IDS groups (*p* = 0.002)On multivariate analysis, the performance of IDS within 4 weeks after NACT and optimal debulking were independent factors for both PFS and OS

**What is the implication of the main finding?**
Performing IDS early after NACT proved to be a good prognostic factor among ovarian cancer patientsMultidisciplinary coordination is required so as to avoid any unnecessary delays

**Abstract:**

Background: There is limited data on the optimal time interval between the last dose of neoadjuvant chemotherapy (NACT) and interval debulking surgery (IDS) in high-grade serous ovarian carcinoma (HGSC). Methods: We retrospectively identified patients with stage IIIC/IV HGSC who received NACT followed by IDS during a 15-year period (January 2003–December 2018) in our Institution. Results: Overall, 115 patients with stage IIIC/IV HGSC were included. The median age of diagnosis was 62.7 years (IQR: 14.0). A total of 76.5% (88/115) of patients were diagnosed with IIIC HGSC and 23.5% (27/115) with IV HGSC. Median PFS was 15.7 months (95% CI: 13.0–18.5), and median OS was 44.7 months (95% CI: 38.8–50.5). Patients were categorized in groups according to the time interval from NACT to IDS: <4 weeks (group A); 4–5 weeks (group B); 5–6 weeks (group C); >6 weeks (group D). Patients with a time interval IDS to NACT ≥4 weeks had significantly shorter PFS (*p* = 0.004) and OS (*p* = 0.002). Median PFS was 26.6 months (95% CI: 24–29.2) for patients undergoing IDS <4 weeks after NACT vs. 14.4 months (95% CI: 12.6–16.2) for those undergoing IDS later (*p* = 0.004). Accordingly, median OS was 66.3 months (95% CI: 39.1–93.4) vs. 39.4 months (95% CI: 31.8–47.0) in the <4 week vs. >4 week time interval NACT to IDS groups (*p* = 0.002). On multivariate analysis, the short time interval (<4 weeks) from NACT to IDS was an independent factor of PFS (*p* = 0.004) and OS (*p* = 0.003). Conclusion: We have demonstrated that performing IDS within four weeks after NACT may be associated with better survival outcomes. Multidisciplinary coordination among ovarian cancer patients is required to avoid any unnecessary delays.

## 1. Background

Ovarian cancer is the eighth leading cause of cancer mortality among women, accounting for over 200,000 deaths in 2020 worldwide [1]. Although most patients initially respond to platinum-based chemotherapy, the majority of patients eventually relapse, and only 25% of patients with stage III/IV disease remain alive at five years [2].

Primary cytoreductive surgery (PDS), followed by platinum-based chemotherapy, is the current standard of care for advanced ovarian cancer. However, for those unsuitable for optimal debulking with no residual tumor, neoadjuvant platinum-based chemotherapy (NACT) for 3–4 cycles followed by interval debulking surgery (IDS) constitutes a valuable alternative. Four phase III trials have also provided evidence that NACT followed by IDS is a non-inferior approach for patients also suitable for PDS [3,4,5,6]. Platinum-based combination therapy with paclitaxel is the preferred regimen for NACT [3,4,5,6]. The addition of bevacizumab to the NACT regimen is feasible and safe according to the GEICO 1205/NOVA and ALTHALYA trials, but showed no difference in complete macroscopic response (CR) rates or progression-free survival (PFS) [7,8].

Delays in adjuvant chemotherapy initiation or surgery post-neoadjuvant chemotherapy are associated with impaired survival in several neoplasms. In ovarian cancer, large retrospective studies have indicated an association between the timing of adjuvant treatment and survival [9,10,11,12]. In addition, delays in postoperative treatment initiation (more than six or seven weeks after IDS) were associated with poor prognosis, especially in patients with no residual disease after surgery [9,10]. However, data regarding the optimal timing of IDS after NACT remains limited, and previous studies have not taken into consideration the molecular biology of the disease. Recent phase III trials recommend performing IDS within six weeks after NACT completion [13,14]. In clinical practice, though, reasons related either to the patient, such as hematological toxicity and performance status, or to the health system and the timely scheduling of the surgical procedure may result in IDS delays. Taking into account that the percentage of advanced ovarian cancer patients treated with NACT constantly increases, the accumulation of data regarding the significance of the ideal time frame within which IDS should be performed is necessary.

Considering the lack of data regarding the optimal interval between the last dose of NACT and IDS, we conducted this retrospective analysis of patients with high-grade serous ovarian cancer (HGSC) of advanced stage (IIIC/IV) treated with NACT and IDS in the Oncology Department of Alexandra University Hospital. We aim to define the impact of delaying IDS after NACT on overall prognosis by using a cutoff of four weeks as defined by previous data.

## 2. Methods

We retrospectively identified patients with stage IIIC/IV ovarian/fallopian tube/primary peritoneal cancer who had received NACT followed by IDS during a 15-year period (January 2003–December 2018) in our institutional database. Our institution has been certified by the European Society of Gyenocologic Oncology (ESGO) as a center of excellence for the treatment of ovarian cancer. The study has been performed in accordance with the 1964 Helsinki Declaration and has been approved by the Institutional Review Board of Alexandra University Hospital (Protocol Number: 513/15-07-2020). Patients were selected for NACT and IDS if it was judged by the experienced gynecologic oncologists that they could not be debulked upfront with no residual tumor. Assessment involved imaging studies and/or laparoscopic evaluation. All subjects received 3 cycles of NACT with carboplatin and paclitaxel according to existing guidelines. All women had provided informed consent for their treatment as well as for the use of their medical records for research purposes. Clinicopathological characteristics, including age at diagnosis, stage, histology, grade, debulking status, BRCA mutation status, type of chemotherapy administered, progression of disease, and overall survival, were collected from the medical records of the patients. Optimal debulking was defined as a maximum residual tumor of <1 cm in diameter after IDS.

### Statistical Analysis

Continuous variables were summarized with the use of descriptive statistical measures [median (IQR; 25–75)], and categorical variables were displayed as frequency tables (N, %). The outcome of the debulking surgery was classified as optimal (residual disease below 1 cm) or suboptimal. Overall survival (OS) was defined as the time between the start of chemotherapy and the date of death from any cause. Progression-free survival (PFS) was defined as the time between the start of chemotherapy and the date of progression. Alive patients were censored at the date of last contact. Kaplan-Meier estimates were used to describe and visualize the effect of categorical variables on OS and PFS. For the analysis, patients were divided into two groups according to the time interval between NACT and IDS: the short (<4 weeks) and long (≥4 weeks) interval groups [15,16,17]. The association of these factors with PFS and OS was assessed through HRs and their 95% confidence intervals estimated from univariate Cox proportional hazard models. Interactions between covariates and time-varying effects were studied. All statistical analyses were performed using SPSS 24.0 statistical software.

## 3. Results

### 3.1. Study Population

Overall, 115 patients with HGSC stage IIIC/IV that underwent NACT followed by IDS were included in our analysis (Table 1). The median age of diagnosis was 62.7 years (IQR: 58.1–71.8). A total of 76.5% (88/115) of patients were diagnosed with stage IIIC HGSC and 23.5% of patients (27/115) with stage IV disease. The result of the debulking surgery was available in 101 patients, among whom 69 (68.3%) were optimally debulked. Performance status was ECOG 0 (53.2%; 59/111) and 1 (34.3%; 38/111) in most cases. Patients were categorized into two groups: 23 patients underwent IDS within 4 weeks from the end of NACT (20%; 23/115), while 92 patients (80%; 92/115) underwent the operation after 4 weeks. The median time interval between the last dose of NACT and IDS was 5.6 weeks (IQR: 4.1–7.0). Specifically, IDS was performed within 4 weeks from NACT completion in 23 patients (20%; 23/115), between weeks 4 and 5 in 20 patients (17.4%; 20/115), between weeks 5 and 6 in 22 patients (19.1%; 22/115), and after week 6 in 50 patients (43.5%; 50/115). BRCA1/2 mutation status was known in 79 patients. A total of 22.8% (18/79) of patients harbored BRCA1/2 somatic mutations, whereas 77.2% (61/79) were BRCA1/2 wild-type. Overall, 81.7% of patients (94/115) had experienced disease progression until the time of the analysis, while 53% (61/115) were deceased. Median PFS was 15.7 months (95% CI: 13.0–18.5) (Appendix A), and median OS was 44.7 months (95% CI: 38.8–50.5) (Appendix A).

### 3.2. Subgroup Analysis

Patients were categorized in groups according to the interval from the last dose of NACT to IDS: <4 weeks (group A); ≥4 to <5 weeks (group B); ≥5 to <6 weeks (group C); ≥6 weeks (group D). Table 2 presents the clinicopathological characteristics of patients undergoing IDS within 4 weeks after NACT vs. those undergoing IDS after 4 weeks after NACT. No statistically significant difference was noted for known prognostic characteristics such as age, stage, and performance status among the four groups of patients. In addition, the outcome of the IDS was similar among groups of patients, while the percentage of patients with known BRCA1/2 mutations also did not differ significantly (Table 2). Patients with a long-time interval from NACT to IDS (≥4 weeks) had significantly poorer PFS (*p* = 0.004) and OS (*p* = 0.002) than those in the short interval (<4 weeks). Median PFS was 26.6 months (95% CI: 24.0–29.2) for patients undergoing IDS <4 weeks after NACT vs. 14.4 months (95% CI: 12.6–16.3) for those undergoing IDS later (*p* = 0.004) (Figure 1). Accordingly, median OS was 66.3 months (95% CI: 39.1–93.4) vs. 39.4 months (95% CI: 31.8–47.0) in the <4 week vs. ≥4 week time interval NACT to IDS groups (*p* = 0.002) (Figure 1).

We then evaluated the impact of delaying IDS after NACT (group A: < 4 weeks; group B: ≥4 to <5; group C: ≥5 to <6; group D: ≥6 weeks). Table 3 summarizes the clinicopathological characteristics of the four subgroups (group A, group B, group C, and group D). The median time interval from NACT to IDS was 3.1 weeks (IQR: 3.0–3.7) for group A, 4.4 weeks (IQR: 4.1–4.7) for group B, 5.4 weeks (IQR: 5.1–5.7) for group C, and 7.1 weeks (IQR: 6.4–7.9) for group D. Kaplan-Meier curves of PFS and OS for each of the subgroups are shown in Figure 2. We observe again that both PFS and OS are significantly higher in group A compared to groups B, C, and D. Among the patients undergoing IDS after week 4, no differences were observed in terms of PFS or OS.

### 3.3. Multivariate Analysis

We performed a Cox regression analysis of the factors that influence PFS and OS in HGSC patients undergoing IDS. Initial status of disease (IIIC or IV), debulking status (optimal vs. suboptimal), ECOG performance status, and interval from NACT to IDS (<4 weeks vs. ≥4 weeks) were included in the analysis. Again, the performance of IDS within 4 weeks after the last dose of NACT retained its statistical significance in terms of PFS (*p* = 0.004) and OS (*p* = 0.003). Optimal debulking was also an independent factor in both PFS (*p* = 0.008) and OS (*p* = 0.001) (Table 4).

## 4. Discussion

There is no consensus regarding the optimal timing of interval debulking surgery after the completion of neoadjuvant platinum-based chemotherapy. We investigated the impact of performing IDS early (within four weeks) after neoadjuvant chemotherapy on PFS and OS. The cut-off of four weeks has been proposed by previous studies as the optional time interval for cytoreductive surgery in terms of safety and efficacy [15,16,17]. Indeed, performing the surgery within four weeks after the last dose of NACT improved both PFS and OS in univariate and multivariate analyses in patients with stage IIIC/IV HGSC.

Previous studies have shown that time to initiation of postoperative chemotherapy is significantly associated with OS [10,11,12]. Time off chemotherapy, defined as the time from the last dose of NACT to the initiation of adjuvant chemotherapy, has also been associated with disease prognosis [16,17,18]. Indeed, postoperative complications, including extended resections, wound healing, and bleeding, may often delay the initiation of adjuvant treatment. However, it is thought that chemotherapy should be started as soon as possible after surgery, especially if debulking was suboptimal. Early initiation of chemotherapy may prevent the distant dissemination of the remaining tumor cells and decrease the tumor burden. It is unclear whether these limitations apply to neoadjuvant chemotherapy as well.

There are several factors that could delay the performance of IDS after NACT completion in a real-world setting. First of all, patients should have recovered from treatment-related toxicities, especially hematologic toxicity, so that surgical intervention can be safely performed. This is of utmost importance for older patients with comorbidities who do not tolerate NACT well. In addition, patients should perform laboratory and imaging tests after chemotherapy completion and be evaluated for their eligibility to undergo IDS by a multidisciplinary team. Finally, logistical reasons related to the availability of operating rooms and the priority lists among ovarian cancer patients in each hospital may also account for delays in performing IDS after NACT.

Data emerging from other solid tumors imply that the time interval between NACT and surgery could affect prognosis [19,20]. In breast cancer, there are studies supporting the idea that breast surgery within three weeks after the completion of NACT would be of maximal benefit [19]. In rectal cancer, a longer time interval between neoadjuvant chemoradiotherapy and surgical excision in locally advanced rectal cancer could lead to higher pCR rates [21]. In non-small cell lung cancer, surgery should not be performed immediately but within a time frame of six weeks after the completion of neoadjuvant chemoradiation [22]. The timing of surgery after neoadjuvant treatment should be carefully balanced in solid tumors to overcome chemotherapy-related toxicities but also to prevent recurrence of residual disease before surgery.

Our data indicate that performing IDS early after NACT (within a month’s timeframe) is an independent prognostic factor for advanced ovarian cancer. Of note, this result seems independent from the biology of the disease since the percentage of BRCA1/2 mutant patients was equally distributed among the groups analyzed. Considering the efficacy of platinum-based chemotherapy in BRCA1/2 mutant patients, one would expect that merely biology would determine prognosis in these patients that receive NACT. However, this is not the case in our cohort of patients. On the other hand, our study result could be attributed to the residual disease post-NACT in ovarian cancer patients. As shown previously by us and others, approximately one third of patients receiving NACT have a very good pathological response to chemotherapy (CRS3) [23,24]. However, this is not equal to pathological complete response (pCR), as detected post-NACT in breast cancer patients. For breast cancer patients with pCR, the timing of surgery may be of minor importance. In ovarian cancer patients, though, delay of IDS may allow expansion of residual disease even in patients with major pathological responses, which could again allow microscopic intraperitoneal dissemination of the disease, hampering the outcome of the debulking surgery. In addition, the four-week timepoint revealed by our data may be considered a statistical cut-off value, but there is also a biological rationale since it coincides with the re-propagation of residual disease post-last chemotherapy cycle.

Our study is characterized by certain limitations. First, this is a retrospective analysis and not a prospective randomized study. In addition, other factors that could influence the overall prognosis were taken into account in the multivariate analysis. However, it is well known that BRCA mutation status is associated with PFS and OS, conferring a survival benefit in BRCA-mutated HGSC patients. Since our analysis goes back to 2003, when BRCA1/2 genetic testing was not a routine test for ovarian cancer patients in Greece, data regarding BRCA1/2 mutation status are limited and therefore not included in the multivariate analysis. In addition, delayed surgery is often the result of a more extensive intraabdominal disease or a worse performance status. Indeed, there are many ranking scores evaluating the likelihood of performing optimal cytoreduction. The peritoneal cancer index (PCI) was developed to evaluate the peritoneal dissemination of intraabdominal or intrapelvic malignant tumors. A PCI score >10 was associated with worse survival in ovarian tumors, even though complete cytoreduction was achieved [25]. Fagotti is a laparoscopy-based score for predicting surgical resectability after evaluating omental, peritoneal, and diaphragmatic involvement, bowel and stomach infiltration, and liver metastases [26]. This score was further modified to identify the patients more likely to achieve optimal cytoreduction [27]. Given the retrospective nature of our study, the evaluation of the disease extent at baseline based on these scores was not feasible. Hence, there is a possible bias toward delaying surgery in patients with an extensive disease spread who would have an adverse prognosis anyway. Finally, this is a single-institution study, including only a small number of patients. Large trials should be designed to address the question of the optimal timing of IDS after NACT.

## 5. Conclusions

To date, the importance of timely surgery after NACT remains under discussion in ovarian cancer. We have demonstrated that performing IDS within four weeks after NACT may be a good prognostic factor. Multidisciplinary coordination among ovarian cancer patients is required so as to avoid any unnecessary delays.

## Figures and Tables

**Figure 1 cancers-15-03519-f001:**
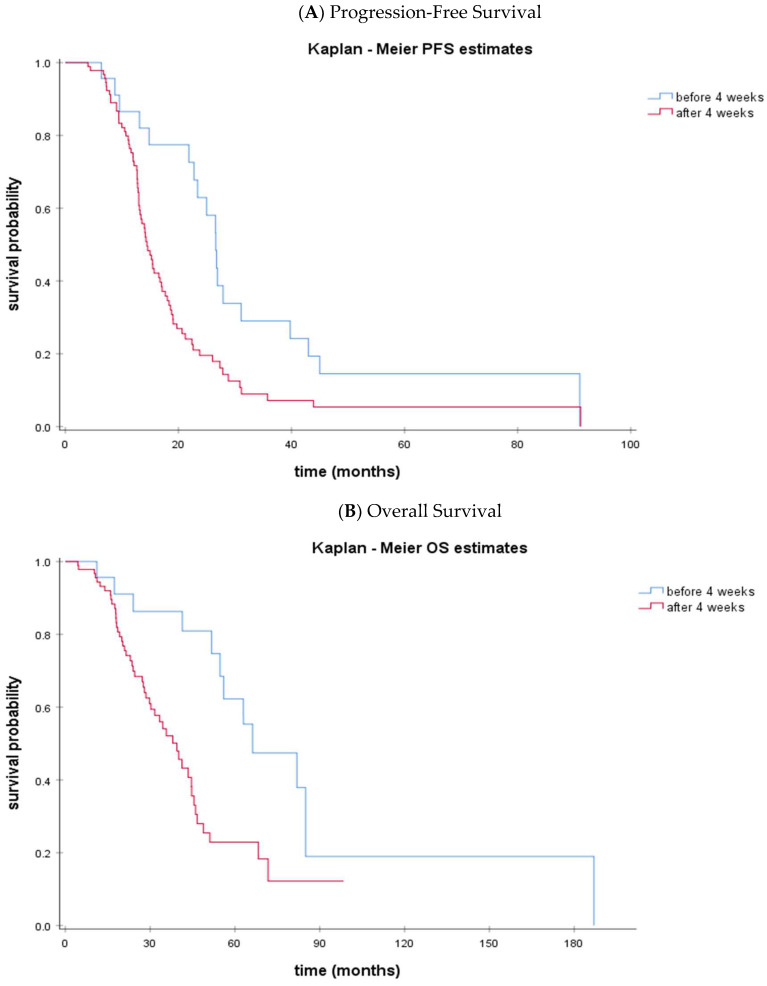
Kaplan-Meier curves of PFS (**A**) and OS (**B**) according to the time interval from NACT to IDS: <4 weeks (blue line) vs. ≥4 weeks (red line).

**Figure 2 cancers-15-03519-f002:**
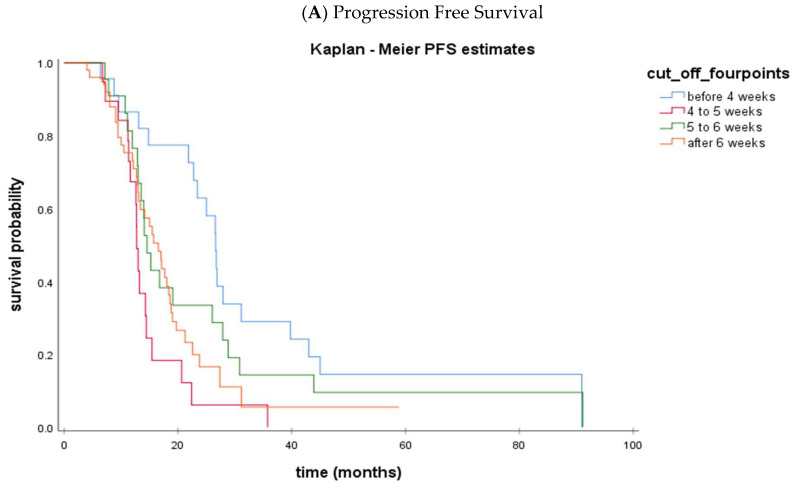
Kaplan-Meier curves of PFS (**A**) and OS (**B**) according to the time interval from NACT to IDS: <4 weeks (Group A) (blue line), ≥4 to <5 weeks (Group B) (red line), ≥5 to <6 weeks (Group C) (green line), ≥6 weeks (Group D) (orange line).

**Table 1 cancers-15-03519-t001:** Clinicopathological characteristics of the overall population.

Characteristic	Total N (%)
**Age at diagnosis, median (IQR: 25–75), years**	62.7 (14.0; 58.1–71.8)
**Initial stage (FIGO)**	
IIIC	88 (76.5%)
IV	27 (23.5%)
**Debulking status**	
Optimal	69 (60%)
Suboptimal	32 (27.8%)
Unknown	14 (12.2%)
**ECOG performance status**	
0/1	97 (84.3%)
2/3	14 (12.2%)
Unknown	4 (3.5%)
**Time interval NACT to IDS, median (IQR: 25–75), weeks**	5.6 (2.9; 4.1–7.0)
**Time interval NACT to IDS, weeks**	
<4	23 (20%)
≥4 to <5	20 (17.4%)
≥5 to <6	22 (19.1%)
≥6	50 (43.5%)
**BRCA1/2 somatic mutation**	
YES	18 (15.7%)
NO	61 (53%)
Unknown	36 (31.3%)
**PFS, median (range), months**	15.7 (13.0–18.5)
**OS, median (range), months**	44.7 (38.8–50.5)

**Table 2 cancers-15-03519-t002:** Clinicopathological characteristics of patients performing IDS within 4 weeks of NACT vs. after 4 weeks.

Characteristic	<4 Weeks N (%)	≥4 Weeks N (%)
**Age at diagnosis, median (IQR: 25–75), years**	61.6 (15.0; 57.0—71.8)	64.0 (14.0; 58.2–71.7)
**Initial stage (FIGO)**		
IIIC	18 (78.3%)	70 (76.1%)
IV	5 (21.7%)	22 (23.9%)
**Debulking status**		
Optimal	14 (60.9%)	55 (59.8%)
Suboptimal	8 (34.8%)	24 (26.1%)
Unknown	1 (4.3%)	13 (14.1%)
**ECOG performance status**		
0/1	19 (82.6%)	78 (84.8%)
2/3	3 (13%)	11 (12%)
Unknown	1 (4.3%)	3 (3.3%)
**BRCA1/2 somatic mutation**		
Yes	3 (13%)	15 (16.3%)
No	14 (60.9%)	47 (51.1%)
Unknown	6 (26.1%)	30 (32.6%)
**Interval NACT to IDS, median (IQR: 25–75), weeks**	3.1 (0.7; 3.0–3.7)	6.0 (2.2; 5.0–7.3)
**PFS, median (95% CI), months**	26.6 (24.0–29.2)	14.4 (12.6–16.3)
**OS, median (95% CI), months**	66.3 (39.1–93.4)	39.4 (31.8–47.0)

**Table 3 cancers-15-03519-t003:** Clinicopathological characteristics of patients performing IDS within 4 weeks (Group A), between weeks 4–5 (Group B), between weeks 5–6 (Group C), and after 6 weeks (Group D) following NACT.

Characteristic	<4 Weeks Group A N (%)	≥4 to <5 Weeks Group B N (%)	≥5 to <6 Weeks Group C N (%)	≥6 Weeks Group D N (%)
**Age at diagnosis, median** **(IQR: 25–75), years**	61.6 (15.0; 57.0–71.8)	60.8 (10.0; 56.1–65.8)	61.6 (19.0; 51.8–71.3)	67.0 (17.0; 59.1–76.1)
**Initial stage (FIGO)**				
IIIC	18 (78.3%)	14 (70.0%)	17 (77.3%)	39 (78.0%)
IV	5 (21.7%)	6 (30.0%)	5 (22.7%)	11 (22.0%)
**Debulking status**				
Optimal	14 (60.9%)	11 (55.0%)	16 (72.7%)	28 (56.0%)
Suboptimal	8 (34.8%)	5 (25.0%)	5 (22.7%)	14 (28.0%)
Unknown	1 (4.3%)	4 (20.0%)	1 (4.5%)	8 (16.0%)
**ECOG performance status**				
0/1	19 (82.6%)	19 (95.0%)	17 (77.3%)	42 (84.0%)
2/3	3 (13.0%)	1 (5.0%)	3 (13.6%)	7 (14.0%)
Unknown	1 (4.3%)	0 (0%)	2 (9.1%)	1 (2.0%)
**BRCA1/2 somatic mutation**				
Yes	3 (13.0%)	4 (20.0%)	3 (13.6%)	8 (16.0%)
No	14 (60.9%)	9 (45.0%)	10 (45.5%)	28 (56.0%)
Unknown	6 (26.1%)	7 (35.0%)	9 (40.9%)	14 (28.0%)
**Interval NACT to IDS, median (IQR: 25–75), weeks**	3.1 (0.7; 3.0–3.7)	4.4 (0.6; 4.1–4.7)	5.4 (0.6; 5.1–5.7)	7.1 (1.5; 6.4–7.9)
**PFS, median (95% CI), months**	26.6 (24.0–29.2)	12.8 (12.2–13.3)	14.6 (12.7–16.4)	16.6 (13.9–19.2)
**OS, median (95% CI), months**	66.3 (39.1–93.4)	39.4 (25.4–53.5)	46.6 (38.1–55.2)	34.5 (25.3–43.7)

**Table 4 cancers-15-03519-t004:** Multivariate analyses for PFS and OS using a Cox proportional hazards model.

Variables	Category	PFS		OS	
		HR (95% CI)	*p*-Value	HR (95% CI)	*p*-Value
**Disease stage**	IIIC vs. IV	0.97 (0.85–1.11)	0.689	0.89 (0.75–1.05)	0.175
**Debulking**	Optimal vs. Suboptimal	1.96 (1.19–3.23)	*0.008*	3.09 (1.63–5.87)	*0.001*
**Performance status**	0/1 vs. 2/3	1.41 (0.67–2.95)	0.370	2.35 (0.99–5.60)	0.053
**Time interval NACT to IDS**	<4 weeks vs. ≥4 weeks	2.33 (1.31–4.17)	*0.004*	3.23 (1.48–7.05)	*0.003*

## Data Availability

Data presented in our study can be found in the patients’ archives that are safely stored in our Institution. The datasets generated during the current study are available from the corresponding author upon request.

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
