# Peer review of "Optimal Time Interval between Neoadjuvant Platinum-Based Chemotherapy and Interval Debulking Surgery in High-Grade Serous Ovarian Cancer"

_cancers, 2023, doi:10.3390/cancers15133519_

Round 1

Reviewer 1 Report

In this manuscript, Andrikopoulou et al, aimed to define the impact of delaying IDS after NACT on overall prognosis of ovarian cancer by using a cutoff of four weeks as defined by previous data. They found that a time interval of NACT to IDS<4 weeks was significantly associated with a prolonged PFS and OS, and performing IDS within 4 weeks after NACT is an independent prognostic factor in advanced HGSC. The research is properly designed, and the findings showed certain novelty and clinical significance. There are some flaws and issues that need further clarification.

1.     The manuscript is not well written, with a lot of typographical and grammar mistakes. The whole flow is really confused and hard to follow. A thoroughly improvement in language editing is needed.

2.     Please make the abbreviations and expression of digit consistent through the manuscript. For example, sometimes it is “four weeks”, and sometimes it is “4 weeks”.

3.     The “Research Highlights” need improvement.

4.     Line 49, “Ovarian cancer is the eight leading cause of cancer mortality” should be “eighth”.

5.     Line 52, “survival rate is only 25% for patients with stage III/IV disease”, please reword.

6.     Line 67, “adverse” should be “poor”.

7.     Line 69, “remain” should be “remains”.

8.     Line 220-222, please reword.

9.     Line 232-240, in this paragraph the authors try to discuss on the factors that “delay the performance of IDS after NACT”. However, the way they presented is telling about factors “influence” the time interval.

The manuscript is not well written, with a lot of typographical and grammar mistakes. The whole flow is really confused and hard to follow. A thoroughly improvement in language editing is needed.

Author Response

REVIEWER 1

In this manuscript, Andrikopoulou et al, aimed to define the impact of delaying IDS after NACT on overall prognosis of ovarian cancer by using a cutoff of four weeks as defined by previous data. They found that a time interval of NACT to IDS<4 weeks was significantly associated with a prolonged PFS and OS, and performing IDS within 4 weeks after NACT is an independent prognostic factor in advanced HGSC. The research is properly designed, and the findings showed certain novelty and clinical significance. There are some flaws and issues that need further clarification.

  1. The manuscript is not well written, with a lot of typographical and grammar mistakes. The whole flow is really confused and hard to follow. A thoroughly improvement in language editing is needed.
  2. Please make the abbreviations and expression of digit consistent through the manuscript. For example, sometimes it is “four weeks”, and sometimes it is “4 weeks”.
  3. The “Research Highlights” need improvement.
  4. Line 49, “Ovarian cancer is the eight leading cause of cancer mortality” should be “eighth”.
  5. Line 52, “survival rate is only 25% for patients with stage III/IV disease”, please reword.
  6. Line 67, “adverse” should be “poor”.
  7. Line 69, “remain” should be “remains”.
  8. Line 220-222, please reword.
  9. Line 232-240, in this paragraph the authors try to discuss on the factors that “delay the performance of IDS after NACT”. However, the way they presented is telling about factors “influence” the time interval.

We would like to thank the Reviewer for his/her positive feedback. We are more than grateful for the time the Reviewer devoted to carefully review our work. We have meticulously examined our manuscript and corrected all typos identified. We have made the modifications indicated by the Reviewer in Lines 60, 63, 78, 80, 230-232 and highlighted in background yellow color.

We would like to thank the Reviewer for his/her thoughtful comment on paragraph Line 232-240. Indeed, we are discussing the factors that delay the performance of IDS after the completion of NACT and thus influence the NACT-IDS time interval. This is exactly the point that we try to raise in this paragraph. If the Reviewer thinks of any other reason that could influence this time interval and deserves mentioning, please inform us to update the context accordingly.

Reviewer 2 Report

Review

OPTIMAL TIME INTERVAL BETWEEN NEOADJUVANT 2 PLATINUM-BASED CHEMOTHERAPY AND INTERVAL 3 DEBULKING SURGERY IN HIGH-GRADE SEROUS OVAR- 4 IAN CANCER

In this retrospective study on 125 patients, the authors evaluate the prognostic importance of the time elapse between the day of the last of 3 courses of neoadjuvant chemo (NSCT) and the day for interval debulking surgery (IDS). They found that a time interval of 4 weeks or more was linked to poorer survival (PFS and OS).

Comments:

This is a well written manuscript on a relevant topic. The authors are to be congratulated for this achievement.

This study is based on a small patient population, which the authors correctly address.

Specific comments:

Page 6, line165-166: “4.4 165 weeks (IQR: 4.1 – 4.7),” for group B is missing

Page 8, line 209-213:

“Again, performance of IDS within 4 weeks after the last dose of NACT retained its statistical significance in terms of PFS (p= 210 0.004) and OS (p= 0.003). Optimal debulking and good performance status were also inde-pendent factors of PFS (all p< 0.05) and optimal debulking was an independent factor of OS (p= 0.001) (Table IV).”

According to Table IV time interval NACT to IDS and optimal debulking had independent prognostic significance for PFS and OS. Disease stage and performance status had not. Must be corrected.

Page 8 line 219-220: “The cut-off of 4 weeks has been proposed by previous studies as the optional time interval to cytoreductive surgery in terms of safety and efficacy”

Needs references

Author Response

REVIEWER 2

This is a well written manuscript on a relevant topic. The authors are to be congratulated for this achievement.

This study is based on a small patient population, which the authors correctly address.

We would like to thank the Reviewer for his/her positive feedback. We are more than grateful for the time the Reviewer devoted to carefully review our work. Here is our point-by-point response to the Reviewer’s comments.

Specific comments:

Page 6, line165-166: “4.4 165 weeks (IQR: 4.1 – 4.7),” for group B is missing

We have modified accordingly the sentence Line 177.

Page 8, line 209-213:

“Again, performance of IDS within 4 weeks after the last dose of NACT retained its statistical significance in terms of PFS (p= 210 0.004) and OS (p= 0.003). Optimal debulking and good performance status were also independent factors of PFS (all p< 0.05) and optimal debulking was an independent factor of OS (p= 0.001) (Table IV).”

According to Table IV time interval NACT to IDS and optimal debulking had independent prognostic significance for PFS and OS. Disease stage and performance status had not. Must be corrected.

We thank the Reviewer for pointing this out. Indeed, only time interval NACT to IDS and optimal debulking were significantly associated with PFS and OS so the text was rephrased accordingly (Line 222-223).

Page 8 line 219-220: “The cut-off of 4 weeks has been proposed by previous studies as the optional time interval to cytoreductive surgery in terms of safety and efficacy”

Needs references

We thank the Reviewer for his/her thoughtful comment. We are now providing three references supporting this cutoff:

1.     “Impact of the Time Interval from Neoadjuvant Chemotherapy to Surgery in Primary Ovarian, Tubal, and Peritoneal Cancer Patients” Chen M. et al, J Cancer. 2018 Oct 18;9(21):4087-4091. doi: 10.7150/jca.26631.

The authors found that “delay in surgery after NACT >4 weeks was associated with a shorter progression-free (P=0.002) but not overall survival (P=0.231)”.

  1. “Impact of the time interval from completion of neoadjuvant chemotherapy to initiation of postoperative adjuvant chemotherapy on the survival of patients with advanced ovarian cancer” Lee Y.J. et al, Gynecol Oncol. 2018 Jan;148(1):62-67. doi: 10.1016/j.ygyno.2017.11.023

They proposed an ideal time interval from NACT to postoperative chemo of 42 weeks (6 weeks) and supported that “patients with longer times to surgery (NACT to IDS) (>25 days; 4 weeks) had significantly poorer OS (P = 0.026) than those with a shorter time to surgery (≤25 days; 4 weeks).

3.      “Choosing the right timing for interval debulking surgery and perioperative chemotherapy may improve the prognosis of advanced epithelial ovarian cancer: a retrospective study”. Wang D. et al, J Ovarian Res. 2021 Mar 27;14(1):49. doi: 10.1186/s13048-021-00801-4

The authors establish a threshold of 35 days (5 weeks) between preoperative and postoperative chemotherapy. They state that: “we usually perform IDS approximately 3 weeks after the last cycle of NACT and start the first postoperative chemotherapy approximately 7 days after IDS”.

All these references were incorporated both in Methods (Line 122) and the Discussion section (Line 230). We are now providing evidence supporting the timeframe we have selected to classify our patients.

Reviewer 3 Report

In this study, Dr Andrikopoulou et al aimed to evaluate if delayed IDS was associated to worse prognosis. 

Please refrain from including the "ESGO Excellence Centre" consideration from the abstract section. 

It is stated that "The outcome of the debulking surgery was classified as optimal (residual 104 disease below 1cm) or suboptimal ". Did the authors consider to register "complete resection", as no macroscopic residual tumor as an outcome? Please justify. 

Please explain why a 4 cycles threshold was considered for the analysis. In the discussion section, it is stated that "The cut-off of 4 weeks has been proposed by previous studies as the optional 219 time interval to cytoreductive surgery in terms of safety and efficacy ". Please quote those articles and specify both in the Methods and the Discussion sections.

Almost a quarter of the patients submitted to surgery presented ECOG 2/3? Was ECOG assessed at the time of the diagnosis or before surgery? Patients with ECOG 2-3 are not usually fit for surgery. Please explain. 

It is stated that "Optimal debulking and good performance status were also independent factors of PFS (all p< 0.05) and optimal debulking was an independent factor of OS (p= 0.001)". This sentence does not match with the results showed at Table 4 (ECOG is not significant). Please rephrase. 

In the discussion section, "In breast cancer, there are studies sup-242 porting that breast surgery within three weeks after the completion of NACT would be of 243 maximal benefit [18]. In rectal cancer, a longer time interval between neoadjuvant chemoradiotherapy and surgical excision in locally advanced rectal cancer could lead to 245 higher pCR rates [20]. " Isn't it contradictory?

"Of note, this result seems independent from the biology of the disease since the percentage of BRCA1/2 mutant patients was equally distributed among the groups analyzed. Considering the efficacy ofplatinum-based chemotherapy in BRCA1/2 mutant patients one would expect that biology would merely determine prognosis in these patients that receive NACT. However, 256 this is not the case in our cohort of patients. " In my opinion, the authors are not entitled to conclude this in the light of the analysis performed. An specific analysis on the interaction of BRCA status and surgery timing would be of great interest. Moreover, BRCA status was not included in the multivariate analysis. A comparable proportion of BRCA patients between both groups does not provide for enough evidence to conclude that statement. 

"In 262 ovarian cancer patients though, delay of IDS may allow expansion of residual disease even 263 in patients with major pathological response that could allow again microscopic intraper-264 itoneal dissemination of the disease, hampering the outcome of the debulking surgery. In 265 addition, the four weeks timepoint revealed by our data may be considered as a statistical 266 cut-off value, but there is also a biological rational since it coincides with re-propagation 267 of residual disease post last chemotherapy cycle". Please quote the references on which these statements are based. 

Did the authors consider to subanalyse the OS / PFS results depending on resection status and the delay on surgery by K-M? Please consider a subanalysis with 4 arms:a)  complete resection, <4 cycles; b) suboptimal, <4 cycles; c) complete, >4 cycles, d) suboptimal <4cycles.

I think a major issue to discuss extensively is the possible bias (the elephant in the room, some may say) that delayed surgery often concurs with a more extensive disease and/or worse PS. Was extension of the disease calculated by any means (PCI; Fagotti....), or controlled by surrogate markers as, i.e., Aletti surgical complexity score? Please consider and explain this in the discussion section. 

In the conclusions section, "We have demonstrated that performing IDS within 4 weeks after NACT 281 may be associated with better survival outcomes". I'm afraid this is a strong sentence to conclude after a retrospective, unicentric and partially-analysed study. 

The manuscript requires minor English and scientific editing (Opening sentences with a figure, i.e.). 

Author Response

In this study, Dr Andrikopoulou et al aimed to evaluate if delayed IDS was associated to worse prognosis. 

Please refrain from including the "ESGO Excellence Centre" consideration from the abstract section. 

We have removed the sentence ESGO Excellence Centre as the Reviewer suggested (Line 29-30).

It is stated that "The outcome of the debulking surgery was classified as optimal (residual 104 disease below 1cm) or suboptimal ". Did the authors consider to register "complete resection", as no macroscopic residual tumor as an outcome? Please justify. 

We thank the Reviewer for his/her thoughtful comment. The sample size is relatively small including only 115 patients. Given this small sample size it would be difficult to further subdivide these patients in three subgroups (complete, optimal and suboptimal resection) so we decided to classify them in two groups.   

Please explain why a 4 cycles threshold was considered for the analysis. In the discussion section, it is stated that "The cut-off of 4 weeks has been proposed by previous studies as the optional 219 time interval to cytoreductive surgery in terms of safety and efficacy ". Please quote those articles and specify both in the Methods and the Discussion sections.

We thank the Reviewer for giving us the opportunity to explain this issue. All patients included in our study received 3 cycles of neoadjuvant platinum-based chemotherapy and not 4 cycles. The patients were not divided according to cycles of neoadjuvant chemotherapy they received prior to IDS – they were classified according to the time interval (number of weeks) between the last dose of NACT and IDS. The cutoff of 4 weeks has been proposed by various studies including the ones below:

1.     “Impact of the Time Interval from Neoadjuvant Chemotherapy to Surgery in Primary Ovarian, Tubal, and Peritoneal Cancer Patients” Chen M. et al, J Cancer. 2018 Oct 18;9(21):4087-4091. doi: 10.7150/jca.26631.

The authors found that “delay in surgery after NACT >4 weeks was associated with a shorter progression-free (P=0.002) but not overall survival (P=0.231)”.

  1. “Impact of the time interval from completion of neoadjuvant chemotherapy to initiation of postoperative adjuvant chemotherapy on the survival of patients with advanced ovarian cancer” Lee Y.J. et al, Gynecol Oncol. 2018 Jan;148(1):62-67. doi: 10.1016/j.ygyno.2017.11.023

They proposed an ideal time interval from NACT to postoperative chemo of 42 weeks (6 weeks) and supported that “patients with longer times to surgery (NACT to IDS) (>25 days; 4 weeks) had significantly poorer OS (P = 0.026) than those with a shorter time to surgery (≤25 days; 4 weeks).

3.      “Choosing the right timing for interval debulking surgery and perioperative chemotherapy may improve the prognosis of advanced epithelial ovarian cancer: a retrospective study”. Wang D. et al, J Ovarian Res. 2021 Mar 27;14(1):49. doi: 10.1186/s13048-021-00801-4

The authors establish a threshold of 35 days (5 weeks) between preoperative and postoperative chemotherapy. They state that: “we usually perform IDS approximately 3 weeks after the last cycle of NACT and start the first postoperative chemotherapy approximately 7 days after IDS”.

Almost a quarter of the patients submitted to surgery presented ECOG 2/3? Was ECOG assessed at the time of the diagnosis or before surgery? Patients with ECOG 2-3 are not usually fit for surgery. Please explain. 

We thank the Reviewer for giving us the opportunity to explain this. Performance status refers to the clinical condition of the patients at baseline before the initiation of treatment. Of course, this clinical condition was subsequently improved and thus these patients were eventually fit for surgery.   

It is stated that "Optimal debulking and good performance status were also independent factors of PFS (all p< 0.05) and optimal debulking was an independent factor of OS (p= 0.001)". This sentence does not match with the results showed at Table 4 (ECOG is not significant). Please rephrase. 

We thank the Reviewer for pointing this out. Indeed, only time interval NACT to IDS and optimal debulking were significantly associated with PFS and OS so the text was rephrased accordingly (Line 222-223).

In the discussion section, "In breast cancer, there are studies supporting that breast surgery within three weeks after the completion of NACT would be of maximal benefit [18]. In rectal cancer, a longer time interval between neoadjuvant chemoradiotherapy and surgical excision in locally advanced rectal cancer could lead to higher pCR rates [20]. " Isn't it contradictory?

We thank the Reviewer for this comment. This statement is not contradictory since there are many studies supporting a greater pCR rate and an improved DFS when time interval between neoadjuvant chemotherapy and surgery is longer in rectal cancer. ( “Impact of interval between neoadjuvant chemoradiotherapy and surgery in rectal cancer patients, Met SW. et al, World J. Gastroenterol. 2020 Aug 21; 26(31): 4624–4638.” “Akgun E, Caliskan C, Bozbiyik O, Yoldas T, Sezak M, Ozkok S, Kose T, Karabulut B, Harman M, Ozutemiz O. Randomized clinical trial of short or long interval between neoadjuvant chemoradiotherapy and surgery for rectal cancer. Br J Surg. 2018;105:1417–1425”).

This may be a result of the delayed tumor response to radiotherapy but also of the lower incidence of perioperative complications after a more prolonged time after the completion of radiotherapy. In this point we try to highlight the importance of identifying the optimal time interval between the end of neoadjuvant treatment and surgical intervention which may vary across different solid tumors. In high-grade epithelial ovarian cancer, this time interval is not well defined. 

"Of note, this result seems independent from the biology of the disease since the percentage of BRCA1/2 mutant patients was equally distributed among the groups analyzed. Considering the efficacy of platinum-based chemotherapy in BRCA1/2 mutant patients one would expect that biology would merely determine prognosis in these patients that receive NACT. However, this is not the case in our cohort of patients. " In my opinion, the authors are not entitled to conclude this in the light of the analysis performed. An specific analysis on the interaction of BRCA status and surgery timing would be of great interest. Moreover, BRCA status was not included in the multivariate analysis. A comparable proportion of BRCA patients between both groups does not provide for enough evidence to conclude that statement. 

We would like to thank the Reviewer for his/her excellent suggestion. Indeed, it is well-known that BRCA1/2 mutations confer sensitivity to platinum-based chemotherapy and thus an improved PFS/OS. However, it is clearly stated that BRCA status is unknown in a large percentage of patients (36; 31.3%) as it is clearly seen in Table 1. Therefore, this is the reason BRCA status was NOT included in multivariate analysis. We are now addressing this issue and we include a sentence in the “limitations of the study” section highlighting that BRCA status was not included in the multivariate analysis (Line 281-286).

"In ovarian cancer patients though, delay of IDS may allow expansion of residual disease even in patients with major pathological response that could allow again microscopic intraperitoneal dissemination of the disease, hampering the outcome of the debulking surgery. In addition, the four weeks timepoint revealed by our data may be considered as a statistical cut-off value, but there is also a biological rational since it coincides with repropagation of residual disease post last chemotherapy cycle". Please quote the references on which these statements are based.

Did the authors consider to subanalyse the OS / PFS results depending on resection status and the delay on surgery by K-M? Please consider a subanalysis with 4 arms:a)  complete resection, <4 cycles; b) suboptimal, <4 cycles; c) complete, >4 cycles, d) suboptimal <4cycles.

We thank the Reviewer for his/her excellent suggestion. However, we have not divided the patients according to the number of cycles of neoadjuvant chemotherapy they have received – they were classified according to the number of weeks of the time interval between the end of neoadjuvant chemo and the performance of IDS. So the classification of patients between those receiving > 4 cycles and <4 cycles of chemotherapy does not apply to the context of our manuscript. Moreover, we did not perform a combined analysis according to both time interval and debulking status due to the limited population of the study.

I think a major issue to discuss extensively is the possible bias (the elephant in the room, some may say) that delayed surgery often concurs with a more extensive disease and/or worse PS. Was extension of the disease calculated by any means (PCI; Fagotti....), or controlled by surrogate markers as, i.e., Aletti surgical complexity score? Please consider and explain this in the discussion section. 

We thank the Reviewer for his/her excellent suggestion. We totally agree with the Reviewer that disease extent is a factor of delaying surgical intervention and could also affect prognosis. Thus, we have now included a paragraph in the “limitations section” of the Discussion (Lines 286-298) to highlight this potential bias of our study. We explain that the extension of disease was not evaluated by the existing ranking scores e.g. PCI, Fagotti due to the retrospective nature of our study. We thank once more the Reviewer for pointing this out.

In the conclusions section, "We have demonstrated that performing IDS within 4 weeks after NACT may be associated with better survival outcomes". I'm afraid this is a strong sentence to conclude after a retrospective, unicentric and partially-analysed study. 

We would like to thank the Reviewer for his/her thoughtful comment. We have used the word “may” in this sentence due to the existence of these limitations the Reviewer highlights. However, we have additionally now rephrased the sentence according to the Reviewer’s suggestion (Line 304).
